# Breastfeeding Duration Is Associated with Regional, but Not Global, Differences in White Matter Tracts

**DOI:** 10.3390/brainsci10010019

**Published:** 2019-12-30

**Authors:** Christopher E. Bauer, James W. Lewis, Julie Brefczynski-Lewis, Chris Frum, Margeaux M. Schade, Marc W. Haut, Hawley E. Montgomery-Downs

**Affiliations:** 1Center for Advanced Imaging, Health Sciences Center, West Virginia University, Morgantown, WV 26506, USA; jblewis@hsc.wvu.edu (J.B.-L.); cfrum@hsc.wvu.edu (C.F.); mhaut@hsc.wvu.edu (M.W.H.); 2Rockefeller Neuroscience Institute, West Virginia University, Morgantown, WV 26506, USA; 3Department of Neuroscience, Health Sciences Center, West Virginia University, Morgantown, WV 26506, USA; 4Department of Neuroscience, Sanders-Brown Center for Aging, University of Kentucky, Lexington, KY 40506, USA; 5Department of Psychology, West Virginia University, Morgantown, WV 26506, USA; mmg58@psu.edu (M.M.S.); Hawley.Montgomery-downs@mail.wvu.edu (H.E.M.-D.); 6Department of Neurology, West Virginia University, Morgantown, WV 26506, USA; 7Department of Behavioral Medicine and Psychiatry, West Virginia University, Morgantown, WV 26506, USA

**Keywords:** breastfeeding, cingulum angular bundle, corpus callosum, diffusion tensor imaging, fractional anisotropy, infant feeding methods, magnetic resonance imaging, pediatrics, superior longitudinal fasciculus, white matter

## Abstract

Extended breastfeeding through infancy confers benefits on neurocognitive performance and intelligence tests, though few have examined the biological basis of these effects. To investigate correlations with breastfeeding, we examined the major white matter tracts in 4–8 year-old children using diffusion tensor imaging and volumetric measurements of the corpus callosum. We found a significant correlation between the duration of infant breastfeeding and fractional anisotropy scores in left-lateralized white matter tracts, including the left superior longitudinal fasciculus and left angular bundle, which is indicative of greater intrahemispheric connectivity. However, in contrast to expectations from earlier studies, no correlations were observed with corpus callosum size, and thus no correlations were observed when using such measures of global interhemispheric white matter connectivity development. These findings suggest a complex but significant positive association between breastfeeding duration and white matter connectivity, including in pathways known to be functionally relevant for reading and language development.

## 1. Introduction

While a topic of some controversy, numerous studies have reported that infant breastfeeding is associated with higher intelligence quotient (IQ) scores [1,2,3,4], elevated academic success [5], and better neurocognitive function [6,7] in childhood and later life. Additionally, a strong, positive correlation has been reported between breastfeeding and verbal and language function [3,5,6,8,9]. Despite the numerous reported neurocognitive benefits of breastfeeding, the neurobiological underpinnings of how these effects may be conferred remains largely unknown. A few neuroimaging studies underscore the idea that brain grey matter and white matter tract maturation may reflect a mechanistic outcome, including increased global grey [4,10] and white matter [3], and, enhanced white matter microstructure in a variety of specific pathways in the brain [8,11]. 

Enhanced nutrition has classically been proposed as one potential biological mechanism that may account for the previously-reported associations [11,12,13]. Specifically, human breast milk provides essential fatty acids, including docosahexaenoic acid (DHA) and arachidonic acid (AA), among others, that are important for central nervous system development, yet are neither readily synthesized by the infant nor available in standard infant formulas. In non-human animals, DHA maintains a critical role in cortical maturation, neuronal differentiation, and neuronal arborization [12]. In humans, neuroimaging studies have indicated that children and adolescents who were born pre-term have reduced total white matter volume [14,15], corpus callosum (CC) volume [14,15,16], and cortical grey matter volume [16]. The mothers of children with attention deficit and hyperactivity disorder (ADHD) have reported breastfeeding for shorter durations compared with mothers of children who do not have ADHD [17]. Cumulatively, these studies suggest that nutritional and development factors—among other influences present early in neurodevelopment—may have significant, long-lasting consequences. Breastfed children are shown to have greater grey matter than non-breastfed children, an effect that has been reported in both humans and macaques [4,10]. A previous study supported the notion that total white matter volume was correlated with increased consumption of breastmilk [3], but did not identify individual tracts or pathways. Consequently, in the current study, our first hypothesis was that breastfeeding duration will be positively correlated with total white matter and fractional anisotropy (FA) scores and volumes of specific sub-sections of the corpus callosum in a pediatric sample.

A study assessing the association between breastfeeding duration and myelin water fraction (a metric for comparing water trapped in myelin) in 1–4 year-old toddlers found positive correlations with differences in the bilateral internal capsule, superior orbital-frontal fasciculus, and left superior-parietal lobe, among others; these areas are thought to be involved with executive function and language development [8]. As an extension of this study, we further sought to establish whether an increased duration of breastfeeding was associated with specific white matter pathways, assessed using Diffusion Weighted Imaging (DWI) measures in older children (4–8 years). Because breastfeeding has been associated with elevated language and executive function, our second hypothesis was that the longitudinally running tracts that connect the temporal and parietal areas to the frontal lobe would show significant differences in FA and volume(s) as a function of feeding method (i.e., breastfeeding versus formula). 

To test these hypotheses, we recruited 4–8 year-old children to undergo magnetic resonance imaging (MRI) and obtained retrospective reports of the method(s) that had been used to feed each child when they were an infant from their parents. We additionally collected thorough documentation and controls, including socio-economic status, maternal factors, and sleep scale factors among other behavioral assessment measures (as part of a larger study). Here we focused on examining the associations between breastfeeding duration, or how many months of infant breastfeeding they had received before being completely weaned, white matter volumes, plus fractional anisotropy (FA) measures in nine established white matter pathways in the left and right hemispheres. 

## 2. Materials and Methods

### 2.1. Participants and Protocol

The study sample analyzed in this report was also the control group from a larger investigation of the impact of breastfeeding on the development of sleep-disordered breathing among 4–8 year-old children. This group of community participants, screened specifically to ensure that they did not have symptoms of sleep-disordered breathing, was recruited using advertisements posted in public places and on social media. The study was approved by the West Virginia University Institutional Review Board; parents were administered written informed consent, and children of ages seven or eight years were administered written assent in accordance with the rules of the Declaration of Helsinki of 1975.

A parent or guardian completed a previously validated screening survey with a brief medical history and an assessment of sleep-disordered breathing symptoms [18]. Inclusion into the control group required that the child: (1) show no more than occasional (twice weekly or less frequent) sleep-disordered breathing symptoms (e.g., snoring, apnea, noisy breathing); (2) had not undergone adeno/tonsillectomy; (3) did not have any other major health problem; (4) had no contraindications for MRI scanning. The parent or guardian of each eligible child also completed a survey about the method(s) they had used to feed the child when they were an infant, and a magnetic safety survey, after which the child underwent MRI scanning. During the scan, participants were permitted to watch a video of their choice.

Twenty-two qualifying participants were administered informed consented. Two of these subsequently withdrew from the study and one was excluded due to MRI acquisition issues. For diffusion-weighted imaging (DWI) scans (also see below), two additional participants were excluded due to excessive head motion artifact, leaving *N* = 17 for DWI (specifically diffusion tensor imaging; DTI) analyses. For corpus callosum (CC) segmentation, three participants were excluded due to segmentation failure, leaving *N* = 16 for CC analyses. There were no significant differences in sex, age, or annual income between children whose data were unavailable (or unusable) versus the larger sample. 

### 2.2. Infant Feeding Methods

Prior to MRI scanning, the parent or guardian completed a form describing how their child had been fed as an infant. Initial categories included “exclusively breastfed”, “exclusively formula fed” or “both”, after which information about the specific duration(s) of the method(s) was collected. At the end of the survey, the respondent used a 5-point scale to indicate how confident they were in their ability to remember this information from 5 (completely confident) to 1 (not confident at all); the mean score was 4.4 (0.72), indicating very high confidence. All information was then sealed in an envelope by the participant and opened only after all scans and processing were complete. 

### 2.3. Magnetic Resonance Imaging Scan Procedures

Parents completed a magnetic safety assessment form. To acclimate children to the imaging environment prior to entering the scanning room, each child was first guided through a quiet simulation MRI scanning procedure (“mock scanner”; Model PST-100444; Psychology Software tools, Inc., Sharpsburg, PA, USA). No sedation was administered during either simulation or actual MRI imaging for any participant. During simulation scanning, children were instructed to remain as still as possible while a short story was read to them (~10 min; with parent/guardian present for support) and could be repeated until they were comfortable with the procedure. As incentive for participation, children earned stickers and small toys at each stage of the simulation scanning and actual MRI scanning. At the conclusion of the study, the family was compensated with a gift card.

### 2.4. Anatomical and Diffusion Weighted Scan Acquisition

Images were acquired using a 3.0 Tesla Siemens Verio scanner (Siemens, Erlangen, Germany), using an 8-channel head coil. Whole-brain T1-weighted anatomical MR images were collected using a standard magnetization-prepared rapid gradient echo (MPRAGE) pulse sequence (~5 min). Parameters were 1 × 1 × 1 mm^3^ voxels with repetition time = 1900–2250 ms; echo time = 3–5 ms; flip angle = 9 degrees; and field of view 200–300 × 200–300 mm^2^. Diffusion weighted images (DWI) were collected using diffusion tensor imaging (DTI) pulse sequences, acquired in the axial plane with 2 × 2 × 2 mm^3^ resolution, 20 directions, with 68 total slices (repetition time (TR) = 9.7 s; echo time = 98 ms; flip angle = 90°; and field of view = 230 mm^2^). 

### 2.5. Anatomical Imaging Processing and Corpus Callosum Measurements

All anatomical MPRAGE scans were reconstructed and segmented using FreeSurfer (version 6.0) [19,20]. As part of the default automated segmentation, the central 5 mm of the corpus callosum (CC) was automatically segmented and divided into 5 segments; this was modified to include the central 13 mm of the CC to more thoroughly sample the white matter tracts. All CCs were normalized by estimated Total Intracranial Volume (eTIV) calculated by FreeSurfer to correct for head size [19]. 

One investigator performed feeding-method blinded quality checks and minor editing where required for all CC slices from each participant, while another trained, blinded assistant performed quality checks on a sample of participants’ CCs. Intra-rater interclass correlation coefficient (ICC) of edited volumes was 0.997 (*p* < 0.001) for six randomly selected participants (including participants from the healthy population in this study, and the sleep-disordered breathing population not reported here). The inter-rater ICC was 0.973 (*p* < 0.001), indicating very high reliability. FreeSurfer was also used to calculate estimates of total brain grey and white matters. Because we hypothesized a global white matter effect of breastfeeding, these two measurements were also used in analyses.

### 2.6. DTI Processing

Diffusion-weighted scans were reconstructed and processed using Tracts Constrained by Underlying Anatomy software (TRACULA in FreeSurfer 6.0) which uses global probabilistic tractography to reconstruct 18 (9 bilaterally) 3-dimensional white matter tracts using the anatomical FreeSurfer reconstruction based on MPRAGE scans for constraint and registration [21]. This process has the advantage of complete automation and could be assessed for accuracy upon inspection. 

Fractional anisotropy (FA) scores (from TRACULA) included: (A) standard FA (averaged over the entire pathway); (B) weighted FA (FA scores multiplied by the probability that a given voxel is in the tract and averaged over the entire pathway); (C) center FA (voxels only included if there is a very high probability they are constrained with the tract [conservative estimate] and averaged over the entire pathway). All three methods were considered in this study, with preference given to center FA due to its conservativeness [21]. Head motion was corrected using metrics of translation and rotation reported by TRACULA and treating them as covariates [22]. 

### 2.7. Statistical Analyses

Statistical analyses were conducted using SPSS software (version 24; IBM, Armonk, NY, USA) and a *p* < 0.05 was considered statistically significant. All variables of interest were initially tested for simple Pearson’s correlation coefficients (between breastfeeding duration and CC volumes, and breastfeeding duration and FA scores) for suitability of use in stepwise linear regression (bidirectional deletion for all predictors, with *p* > 0.05 for inclusion and *p* > 0.10 for exclusion), which was used to examine the linear relationship between breastfeeding duration and dependent variables (CC volumes, FA scores). Stepwise linear regression was ultimately not used to test associations between breastfeeding duration and CC volume, as there were no statistically significant bivariate correlations, however stepwise linear regression was used to test correlations between breastfeeding duration and FA scores for all 9 tracts in an exploratory effort. 

No outliers were excluded as none fell more than 3 times the interquartile range from the 1st or 3rd quartile (i.e., there were no outliers within DTI scores at 1.5 times the interquartile range). Age, sex, annual household income, maternal age, maternal education and parental smoking were tested to determine whether they were correlated with either the predictor or dependent variables (only age was positively correlated with CC volume, *p* < 0.05); sex, annual income, and smoke exposure were not significantly correlated but were nonetheless entered into the models as potential covariates in linear regression because of their previously-reported associations with one or both variables [23,24,25]. After the bidirectional removal of non-significant predictors, only breastfeeding duration remained (all confounding variables were ultimately excluded inform the final model). 

Variables were tested for normality using the Shapiro–Wilk method and none were found to violate any assumptions for linear regression. For group comparisons (i.e., those who had been exclusively breastfed compared to those who had been fed any formula), both predictor and dependent variables were tested for normality. Independent *t*-tests were used to compare groups as all variables were normally distributed (except the central portion of the CC, for which the Mann–Whitney *U* test was used). When *t*-tests were statistically significant, analysis of covariance (ANCOVA) was used as a follow-up to control for variance accounted for by relevant confounding variables. 

## 3. Results

Demographics and other characteristics of the 19 participants whose data were included in the analyses are described in Table 1. 

### 3.1. Breastfeeding Duration and CC Volume

Breastfeeding duration was not significantly associated with total CC volume after eTIV correction (*r* = −0.20; *p* = 0.46). Nor were sub-sections significantly associated: anterior CC volume (*r* = −0.31; *p* = 0.24), anterior-central CC volume (*r* = 0.04; *p* = 0.88), central CC volume (*r* = 0.04; *p* = 0.89), posterior-central CC volume (*r* = −0.15; *p* = 0.57), or posterior CC volume (*r* = −0.18; *p* = 0.50). 

There was no significant difference in total CC volume (e.g., Figure 1) between children who were exclusively breastfed compared to those who had received any formula feeding (Table 2; *t* = 0.94; *p* = 0.36). This was also true when looking separately at anterior (*t* = 1.19; *p* = 0.26), posterior (*t* = 0.07; *p* = 0.95), and central (*u* = 31.0; *p* = 0.96) portions. In addition, after correcting for eTIV, there was not a significant association between breastfeeding duration and total white matter (*r* = −0.45; *p* = 0.08) or total grey matter volume (*r* = −0.33; *p* = 0.21). Furthermore, there was no significant difference between groups that were exclusively breastfed versus those that received any formula feeding on either total white matter volume (*t* = 0.61; *p* = 0.55) or grey matter volume (*t* = 0.31; *p* = 0.76).

### 3.2. Breastfeeding Duration and DTI

Using stepwise linear regression, breastfeeding duration was positively associated with center FA scores of the left superior longitudinal fasciculus (SLF) (temporal portion; see Figure 2) in the final model (Figure 3; *p* = 0.019; *R*^2^ = 0.35). However, there were no statistically significant associations between breastfeeding duration and standard FA scores (*p* = 0.14), while a weighted FA score trended toward significance (*p* = 0.054) for the left SLF correction for covariates in the final model.

Breastfeeding duration was positively associated with both standard (*p* = 0.034; *R*^2^ = 0.30) and weighted (*p* = 0.037, *R*^2^ = 0.29) FA scores in left angular bundle (AB) in the final model (Figure 4), but not center FA scores (*p* = 0.85). Finally, children who had been exclusively breastfed when they were infants had significantly greater center FA scores for the left SLF when adjusting for average image translation and rotation, and for sex, age, annual income, and parental smoking, compared to children who had been fed any formula (*F* = 6.9; *p* = 0.034). 

## 4. Discussion

The main study finding was that infant feeding methods are associated with certain left-lateralized white matter pathway changes in the brain, later in development. Specifically, the results indicate that breastfeeding duration is associated with higher fractional anisotropy (diffusivity along a given direction) in the left superior longitudinal fasciculus (SLF) and left arcuate bundle in 4–8 year-old children. Thus, we infer some type of elevated myelination connectivity (or axon diameter, fiber density, or fiber organization) in these areas correlated with breastfeeding duration. 

Previous studies have reported that increased breast milk consumption and breastfeeding duration is correlated with more global measures of white matter volume or quality [3,8,11], which inspired us to extend those findings by identifying the particular tracts that might represent global aspects of white matter tract development. Somewhat surprisingly, however, the main hypothesis of the study, that infant breastfeeding duration would be positively correlated with child corpus callosum (CC) volume, was not supported. The CC was selected as a pathway of interest, in part due to its prominence as the largest white matter tract in the neurotypical brain, its importance in interhemispheric connectivity [26], and implications for this tract in developmental disorders [27,28]. Additionally, we found no evidence of a correlation between breastfeeding duration and CC volume within these data, nor any associations with total white matter volume. Rather, the results suggest a highly specific, left-lateralized association with infant feeding methods. Yendiki and colleagues [21] stated that the left SLF reconstructed using TRACULA most closely resembled the arcuate fasciculus. This is noteworthy in that the superior longitudinal fasciculus—and, in particular, the arcuate fasciculus with its proposed evolutions relative to extant primates—is thought to be heavily involved in human language functions [29,30,31,32], connecting Broca’s and Wernicke’s area (classic model). 

Interestingly, a study investigating language impairment in children with Autism Spectrum Disorders (ASD) found that more severe language impairment is correlated with elevated mean diffusivity of the left SLF, implying that disruption in connectivity from this pathway may result in language deficits [33]. Furthermore, the left SLF (especially the temporal portion) results in the present study support and extend some of the findings by Deoni and colleagues [8], in that they found differences in this tract (using water fraction measures) when comparing a younger cohort of children/toddlers (while sleeping) who were exclusively breastfed compared to those with any formula-feeding. 

We also found an association between breastfeeding duration and FA scores of the left angular bundle (AB). Anatomically, this pathway is believed to connect the hippocampus and entorhinal cortex to the posterior cingulate cortex [34,35]. Functionally, the left AB (in non-demented older adults) is positively correlated with verbal episodic memory performance [35], but there was no association with the right angular bundle, and left or right cingulate bundle. The findings in the present study, that breastfeeding duration may be associated with FA scores in the left AB, are germane to the frequent observations associating breastfeeding with elevated neurocognitive performance [1,2,3,6,7]. Our results imply that infant feeding methods are correlated with two left-lateralized white matter tracts that in humans are primarily implicated in language functioning and verbal episodic memory, two neurocognitive areas that could help explain the respective reported benefits of breastfeeding [3,5,6,8,9]. 

Breastfeeding duration is not the only factor that might predict white matter changes. Non-linear increases in language ability, cognition, and sensorimotor skills also likely influence the development of white matter tracts. Specifically, the introduction of formal education may be a contributing factor as children receive training in the aforementioned areas. Though our results suggest that breastfeeding duration can predict changes in the brain that may affect learning in these areas, differences in quality or duration of education may also account for some of these differences. We attempted to partially address this issue by using age (those who are older have been in school longer) and annual income (those in higher socioeconomic brackets may receive higher quality education), but we cannot fully account for this with the present data. 

There are a number of reasons why we may not have replicated the finding associating breastfeeding duration and CC volume or FA scores, as has been found by others [3,8,10,11]. First, the growth and maturation of both grey and white matter in early development is dynamic and quite different from in an adult brain [11]. Although the developmental growth curves, as measured by neuroimaging, suggest relative stability by 4 years of age [11], we did not model volumetric or structural changes (due to a difficulty with modeling this relatively small sample), which may account for some discrepancies from the literature. Secondly, although we employed stringent quality control protocols (see Section 2.5), we ultimately relied on a semi-automated program protocol for segmentation, which was designed for use in adults rather than children [36]. Nevertheless, we were able to find robust associations in the left SLF and AB, consistent with other neuroimaging studies [8,11]. This study successfully imaged a young population that can be very challenging to scan using MRI [37]. In addition, the methods employed for measuring CC and DTI have been well established and almost entirely automated, and the FA scores were calculated in three different ways. Nevertheless, this study was not without limitations. First, even for a pediatric neuroimaging study, the total sample was relatively small. Thus, we may not have had adequate power to detect changes in some of the more subtle or variable white matter tracts. Second, as reported in the results, the three measures of FA did not always agree. Specifically, breastfeeding duration was associated with left SLF temporal portion center FA scores, but not standard or weighted FA scores, while when examining the left AB, the statistical significance was associated with standard and weighted FA scores but not center FA scores. Concerning the three different measures, the center FA metric was the most conservative, thus there was a higher confidence in the finding that the FA scores in the voxels most likely to be left SLF are indeed correlated with breastfeeding duration, but that this correlation becomes non-significant when voxels less likely to be left SLF are included. Insufficient power might explain the opposite situation of the left AB. Finally, the stepwise linear regression was explored on all nine available major white matter tracts in exploratory fashion; therefore, future studies might focus hypotheses more on left-lateralized longitudinal running tracts such as the SLF and AB. 

In conclusion, the present results indicate a lack of association between infant feeding methods and CC volume, total white matter volume, and total grey matter volume. Instead, the present DWI/DTI evidence indicates that breastfeeding duration is correlated with two distinct, left-lateralized white matter pathways that have each been associated with language function and verbal memory. As numerous studies have indicated that breastfeeding confers benefits in intelligence and academic and neurocognitive functioning, the present results reveal an underlying neural circuitry that might mediate such functions. Thus, these findings also add to the consensus that breastfeeding has a positive effect on brain development.

## Figures and Tables

**Figure 1 brainsci-10-00019-f001:**
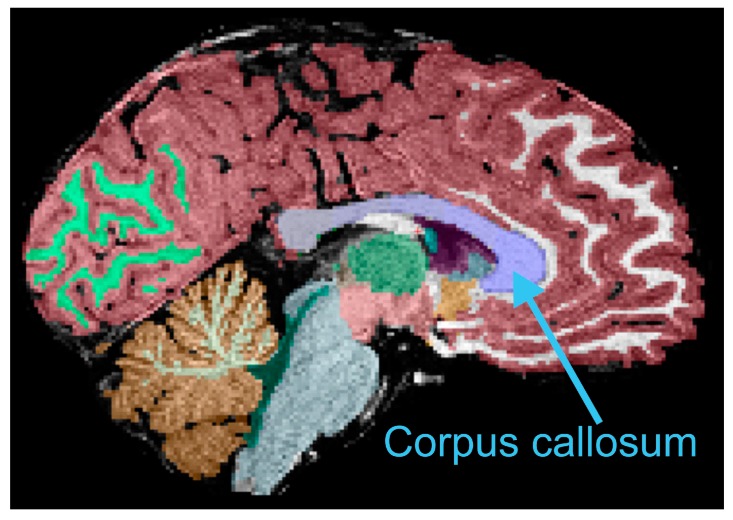
Example of automated segmentation of the corpus callosum (blue). Refer to text for other details.

**Figure 2 brainsci-10-00019-f002:**
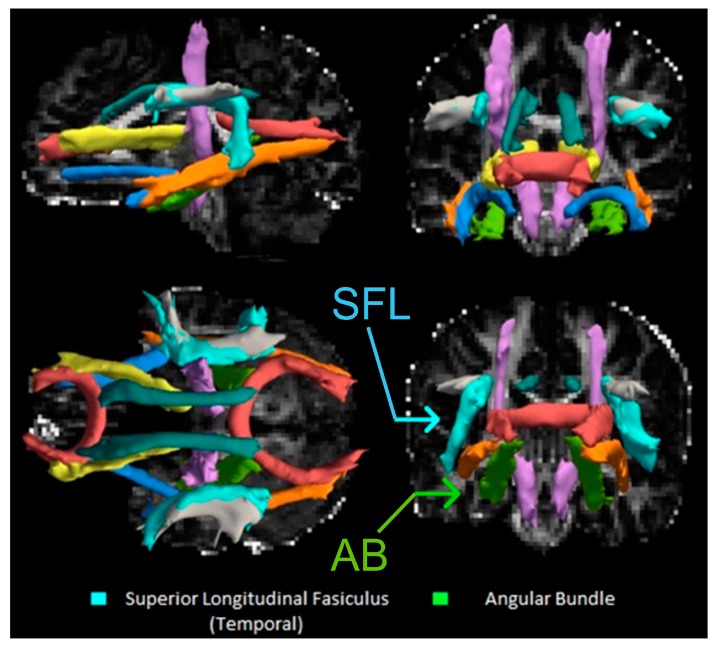
DTI Reconstructions of Selected Pathways Using TRACULA. Top Left: Mid-Sagittal slice of the brain looking at reconstructed pathways. Cyan = superior longitudinal fasciculus (SLF); light green = left angular bundle. Other pathways include the corpus callosum forceps (red), inferior longitudinal fasciculus (orange), corticospinal tract (purple), cingulum bundle (dark teal), anterior thalamic radiation (yellow), SLF-parietal portion (grey), and uncinate fasciculus (blue). Top Right: Coronal slice of the brain looking at the anterior tracts. Bottom Right: Coronal slice of the brain looking at posterior tracts. Bottom Left: Longitudinal slice of the brain with the most superior tracts appearing closer and inferior tracts appearing farther away.

**Figure 3 brainsci-10-00019-f003:**
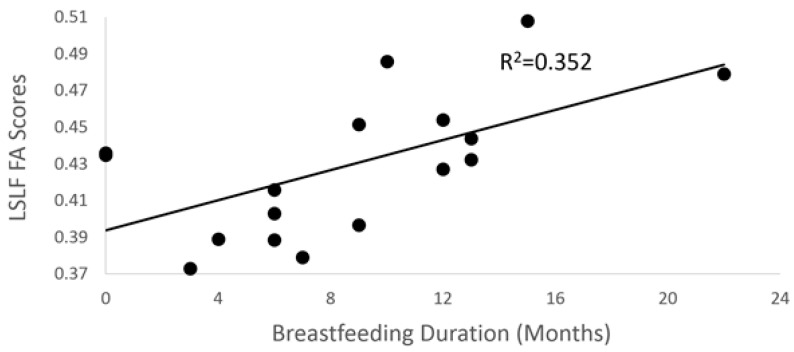
Scatter plot between breastfeeding duration and center FA scores (*p* = 0.019) for the left superior longitudinal fasciculus (SLF).

**Figure 4 brainsci-10-00019-f004:**
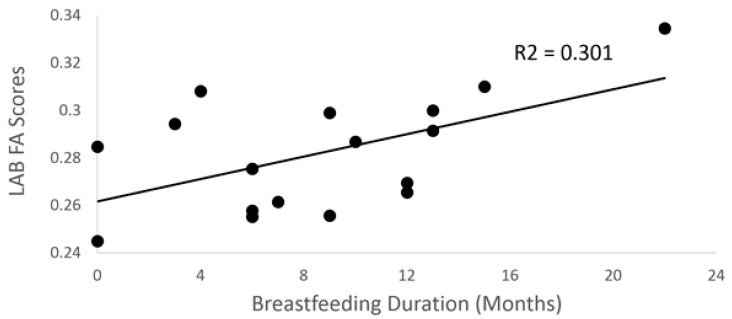
Scatter plot between breastfeeding duration and FA scores (*p* = 0.034) for the left angular bundle.

**Table 1 brainsci-10-00019-t001:** Participant Demographics. ADD = attention deficit disorder.

Characteristic	Value
Age (Years)	6.4 (1.6)
Female (*n*, %)	7 (36.8)
Annual Income (Dollars; Thousands)	92.5 (43) *
Maternal Education (Years)	17.4 (1.9)
Parental Smoking (*n*, %)	3 (15.8)
Diagnosed ADD/ADHD (*n*, %)	1 (5.3)
Exclusive Breastfeeding (*n*, %)	7 (36.8)
Breastfeeding Duration (Months)	9.9 (7.1)
Exclusive Formula Feeding (*n*, %)	2 (10.5)

* Three participants declined to answer.

**Table 2 brainsci-10-00019-t002:** Uncorrected Corpus Callosum Volumes (mm^3^) and eTIV (cm^3^).

Characteristic	Total	Exclusively Breastfed	Any Formula
*n*	16	7	9
Total CC Volume	7353 (790)	7287 (893)	7405 (752)
Anterior CC Volume	2092 (263)	2043 (352)	2130 (181)
Anterior-Central CC Volume	942 (173)	966 (92)	924 (221)
Central CC Volume	948 (130)	955 (117)	943 (147)
Posterior-Central CC Volume	1200 (244)	1114 (198)	1266 (266)
Posterior CC Volume	2171 (344)	2208 (258)	2143 (412)
eTIV	1456 (109)	1486 (102)	1433 (114)

Reported in means and standard deviation. All values are reported in means and standard deviation. Uncorrected corpus callosum values are shown to compare groups in meaningful units (mm^3^). Corrected values can be obtained by dividing by the estimated intracranial volume (eTIV).

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
