# Peer review of "Breastfeeding Duration Is Associated with Regional, but Not Global, Differences in White Matter Tracts"

_brainsci, 2019, doi:10.3390/brainsci10010019_

Round 1

Reviewer 1 Report

Dear authors,

Thank you for the opportunity to review your manuscript entitled "Breastfeeding Duration is Associated with Regional but Not Global, Differences in White Matter Tracts." Here you report a positive relationship between breastfeeding duration and white matter integrity, as quantified by FA, within several major white-matter tracts, including the left SLF and the left angular bundle. The suggestion from your findings is that previously observed relationships between breastfeeding duration and language/general intelligence may, in part, be driven by white-matter development related to breastfeeding.

Overall, I found the background information to be relevant and appropriate to the questions being asked. Likewise, I found the discussion and contextualization of the results to be appropriate. However, there are several aspects of the pre-processing and statistical analyses that should be clarified to allow for reproducibility and context. These are detailed below

Bivariate correlations were assessed for suitability for inclusion the stepwise regression (line 162-165). What was the criterion for inclusion? Forward, backward, or bidirectional deletion? Relatedly, most potential confounds were ultimately included regardless of their relationship (only age was associated with CC volume) on the basis of extant literature. Therefore, what was the purpose of using bivariate relationships to assess suitability? Further, were all variables retained after stepwise variable selection? If so, please comment on the effects of these predictors. My specific concern is that there are a large number of independent variables (7 independent variables, including the one of interest) measured on 16 participants (roughly twice as many observations as variables). If not all the variables were retained, please indicate which were dropped, as the current description suggests that all were retained.  Were R2 values as reported the model R2 or partial R2? In light of the large number of predictors, it would be helpful to know what the overall model fits were.  Were all of the TRACULA tracts evaluated or only the SLF/AB? If all tracts were included, what was done for multiple comparisons correction? If only the left SLF/AB were considered, this needs to be more substantially justified in the introduction and hypotheses.

With clarifications regarding these concerns, I find that the manuscript would be substantially clarified to ensure, as noted above, reproducibility and contextualization.

Thank you again for the opportunity and I look forward to reading a revision.

Author Response

Please see attachment, and specific reference to "Reviewer 1" comments in dark blue, indented text.

Reviewer 2 Report

This pediatric MRI study investigated how recalled duration of infant breastfeeding could be reflected in grey matter and white matter measures in children in the age range of 4-8 years old. Correlational analyses showed significant impact of breastfeeding duration on left-lateralized white matter tracts. However, significant correlations were not observed between breastfeeding duration and measures of interhemispheric connectivity including corpus callosum size.

The overall conceptual framework for the work is well defined. The text is well written. But there are some theoretical and technical issues that need to be addressed before I would recommend acceptance for publication.

The literature review needs to be reflect all the current studies on the topic. Likewise, the discussion section needs highlight the similarities across the studies as well as the differences and possible explanations for the differences. There needs to be some serious consideration about the developmental trajectory of grey matter and white matter changes early in life. A child's brain structure undergoes rapid development in the first 6 years and is very different from the adult brain. Direct application of adult-based software packages such as Freesurfer and its plugin packages have been shown to produce inaccurate estimates (See Phan et al., 2018 for a review). This analytical issue needs to be taken into account. In addition to early influences from breastfeeding, multiple co-varying factors need to be taken into account. In particular, language ability (including reading), cognitive ability, and sensorimotor skills undergo dramatic changes early in life. The age range of 4-8 includes preschool and school age children, and how the anatomical measures might be influenced by the introduction of formal education (including reading, second language learning, musical training, etc.) is not taken seriously in the discussion. Exactly how the correlation analysis was implemented appears poorly described since there were so many independent variables and dependent variables. For instance, were there correlations among the residuals in your multiple models? Was there correction for multiple comparison (for example, correction with false discovery rate)? Could mixed effect models be a better approach for the multivariate analysis by taking into account random variables and the potential mutual dependencies among the variables (some of which are likely not independent)? 

Some references:

Phan, T. V., Smeets, D., Talcott, J. B., & Vandermosten, M. (2018). Processing of structural neuroimaging data in young children: Bridging the gap between current practice and state-of-the-art methods. Developmental Cognitive Neuroscience, 33, 206-223. doi: https://doi.org/10.1016/j.dcn.2017.08.009

Deoni, S., Dean, D., Joelson, S., O'Regan, J., & Schneider, N. (2018). Early nutrition influences developmental myelination and cognition in infants and young children. NeuroImage, 178, 649-659. doi: https://doi.org/10.1016/j.neuroimage.2017.12.056

Liu, Z., Neuringer, M., Erdman, J. W., Jr., Kuchan, M. J., Renner, L., Johnson, E. E., . . . Kroenke, C. D. (2019). The effects of breastfeeding versus formula-feeding on cerebral cortex maturation in infant rhesus macaques. NeuroImage, 184, 372-385. doi: 10.1016/j.neuroimage.2018.09.015

Luby, J. L., Belden, A. C., Whalen, D., Harms, M. P., & Barch, D. M. (2016). Breastfeeding and Childhood IQ: The Mediating Role of Gray Matter Volume. J Am Acad Child Adolesc Psychiatry, 55(5), 367-375. doi: 10.1016/j.jaac.2016.02.009

Ou, X., Andres, A., Pivik, R. T., Cleves, M. A., Snow, J. H., Ding, Z., & Badger, T. M. (2016). Voxel-Based Morphometry and fMRI Revealed Differences in Brain Gray Matter in Breastfed and Milk Formula–Fed Children. American Journal of Neuroradiology, 37(4), 713-719. doi: 10.3174/ajnr.A4593

Stadler, D. D., Musser, E. D., Holton, K. F., Shannon, J., & Nigg, J. T. (2016). Recalled Initiation and Duration of Maternal Breastfeeding Among Children with and Without ADHD in a Well Characterized Case-Control Sample. J Abnorm Child Psychol, 44(2), 347-355. doi: 10.1007/s10802-015-9987-9

Author Response

Please see attachment, and specific reference to "Reviewer #2" comments in dark blue, indented text.

Round 2

Reviewer 1 Report

Thank you for addressing my concerns. I am satisfied with the explanations and edits.

Reviewer 2 Report

The researchers have addressed my concerns in this revision.